# Learning to Rank Generation with Pairwise Partial Rewards

**Youngwon Lee**[*], **Jinu Lee**[*], **Seung-won Hwang**[†]

Seoul National University

ywlee@ldi.snu.ac.kr, {aquamrn, seungwonh}@snu.ac.kr

## Abstract

This paper studies the use of reinforcement learning for conditional text generation, which overcomes the limitation of the prevalent supervised maximum likelihood estimation approach. However, it still suffers from challenges including the large action space and the delayed reward, as the reward can be computed only after an entire sequence is generated. To address these challenges, we propose a method that provides partial rewards for intermediate actions taken on partial sequences. This enables the model to promptly prioritize actions that lead to the generation of more desirable sequences. Our method's key contribution lies in its focus on distinguishing relatively more desirable actions rather than striving to precisely estimate pointwise values for arbitrary partial sequences. Instead, our reward shaping method learns to discern the relative desirability between pairs of actions, or rank actions in a pairwise manner, only when necessary and feasible. This is materialized in an efficient way by leveraging the prefix tree constructed from the sampled sequences. Experimental results on paraphrase generation and constrained machine translation tasks showcase the effectiveness of our method.[1]

## 1 Introduction

Conditional text generation encompasses various tasks, with the objective of generating a sequence $\mathbf{y}$ based on a given source sequence $\mathbf{x}$, in a way that maximizes the utility $u$. For example, in paraphrase generation, which we use as our running example, $u$ is high when (1) $\mathbf{y}$ matches the meaning of the source sequence $\mathbf{x}$, (2) $\mathbf{y}$ is fluent in terms of language modeling, and (3) it has high lexical dissimilarity from $\mathbf{x}$ (Kumar et al., 2020;

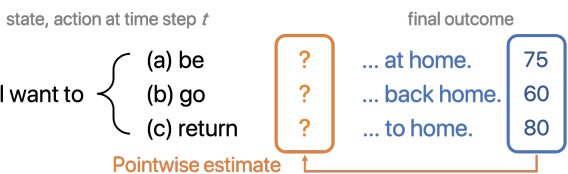

Figure 1: Reinforcement learning boils down to either implicitly or explicitly maintain pointwise estimates of values at each state, corresponding to utility of partially generated sequences.

Cao and Wan, 2020; Nighojkar and Licato, 2021, *inter alia*).

Supervised learning, particularly maximum likelihood estimation (MLE) training, has typically been the standard approach for conditional text generation. In this approach, the model is trained to maximize the likelihood of the reference given the source. During inference, it generates output sequences for which it assigns high probability.

This approach is known to suffer from two significant challenges: exposure bias and train-test objective discrepancy (Ranzato et al., 2016). To address both challenges, reinforcement learning (RL) has gained attention as an alternative (Shen et al., 2016; Kreutzer et al., 2017). RL formulates autoregressive text generation as a Markov decision process where a *state* corresponds to a generated prefix under construction,[2] an *action* represents the selection of the next token, guided by the policy network, which is essentially the language model. The agent's goal is to maximize the reward it receives, which is determined by the utility $u$ of the output sequence.

However, this utility can be computed only after the generation is finished. This sparse and delayed reward problem introduces challenges in learning the impact of each action taken by the model, known as the *credit assignment problem* in RL, as depicted in Figure 1 (Guo et al., 2022; Li

---

[*]Equal contribution.

[†]Corresponding author.

[1]Our code is publicly available at: https://github.com/jinulee-v/PairwisePartialReward

[2]Or, defined to include the source, $s_t = (\mathbf{x}, \mathbf{y}_{\leq t})$.

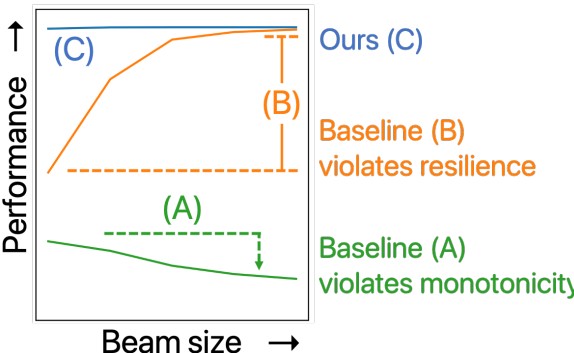

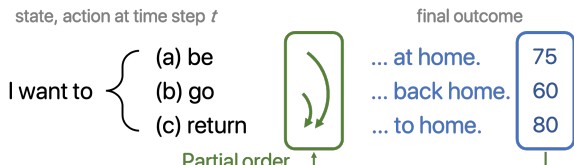

Figure 3: The proposed PPR *learns to rank* actions, defining partial (rather than total) order over partial sequences. The model receives partial reward, instead of waiting for the delayed reward.

Figure 2: (*Beam monotonicity* and *resilience*) Baselines fall into either of the failure cases: (A) fails to achieve monotonicity while (B) exhibits monotonicity but fails at resilience. Our method (C) achieves both, retaining the performance even with smaller beam sizes.

et al., 2017). When a pointwise estimation of each partial candidate is unreliable, generation leads to undesirable sequences such as blank or repeated $n$-grams (Holtzman et al., 2020).

We hypothesize that such pointwise estimation of state values, or equivalently expecting models to predict a *total order* over all partially generated sequences, is an overly ambitious objective. In this sense, while text degeneration is often attributed to the failure of decoding, we show that the policy, the model itself, is more responsible to the failure.

To support our claim that language models indeed struggle with the pointwise estimation of the final outcome from partially generated sequences, we suggest comparing their performance with beam search using different beam sizes, as illustrated in Figure 2. Beam search can be seen as providing some tolerance for the model's mistakes by holding on to candidate sequences even when its prefix values are underestimated (Meister et al., 2020). Ideally, if the model's belief about the sequence's utility aligns positively with the actual utility of the fully generated sequence, increasing the beam size should only increase such a tolerance. We refer to this property as *beam monotonicity*, which is violated in a baseline model shown in Figure 2A.

Similarly, reducing the beam size should not significantly hurt the performance if the model accurately estimates values of prefixes in earlier stages, as higher-ranked candidates would still survive with a smaller beam size. We call this property *beam resilience*, which is violated in Figure 2B. A reliable policy should satisfy both properties, as shown in Figure 2C.

Our distinction is proposing a reliable policy in

Figure 2C, by adjusting an overly ambitious requirement of full ordering, to partially ordering a subset of possible pairs of partial sequences. Essentially, we aim to train the model to choose actions that will lead to better outcomes, aligning with our training objective. This approach also provides the model with partial rewards, offering direct feedback on individual actions long before the delayed reward becomes available.

Our method, called PPR (Pairwise Partial Reward), implements this idea by utilizing a prefix tree constructed from the sampled sequences. PPR employs a pairwise ranking objective for the branching nodes in the prefix tree. Firstly, we assess the desirability of following each branch by considering the leaf nodes in the corresponding subtree and their associated observed utilities. Then, we compare pairs of branches and encourage the model to prefer the branch with a higher expected return, effectively establishing a partial order over partially generated sequences. For example, in Figure 3, PPR can encourage the model to prefer action (or the resulting partial sequence) $c$ over $a$ or $b$, defining ordering of $a \rightarrow c$ and $b \rightarrow c$ as shown in the figure.

Importantly, PPR does not require the model to directly compare its value estimate of $c$ with other partial sequences apart from $a$ or $b$. Additionally, it can be seen as the model receiving partial reward for choosing the action $c$. In summary, this paper makes two contributions:

- We introduce a novel method for reward shaping in conditional text generation through pairwise learning to rank.

- We demonstrate the effectiveness of our approach, showcasing not only high-quality generation but also improved estimation of partial sequence values through beam monotonicity and resilience.

## 2 Background

As a background, we formulate the description of how autoregressive conditional text generation is formulated as a Markov decision process and the agent-environment interaction is defined to transform the task into an RL problem. As mentioned before, state $s_t$ at time step $t$ corresponds to $(\mathbf{x}, \mathbf{y}_{\leq t})$, and the possible action $a_t$ at this moment is the act of choosing the next token, $y_{t+1}$ to reach the next state $s_{t+1}$. Here the environment is fully observable and the state transition given the action is deterministic, that is, selecting an action (the next token) is equivalent to selecting the next state (prefix with length increased by 1). A conditional language model then can be regarded as directly parameterizing the policy, $\pi_\theta(a_t \mid s_t) = p_\theta(y_{t+1} \mid \mathbf{x}, \mathbf{y}_{\leq t})$.

Typically, the interaction with the environment is simulated as the delayed reward

$$R_t = \begin{cases} u(\mathbf{y}; \mathbf{x}), & \text{if } t = |\mathbf{y}|, \\ 0, & \text{otherwise,} \end{cases} \quad (1)$$

where $R_t := R(s_{t-1}, a_{t-1}, s_t)$ is the reward for making transition from $s_{t-1}$ to $s_t$ by choosing the action $a_t$. Delayed reward hinders the model from figuring out which actions in the past should be attributed to the final outcome.

## 3 Method

### 3.1 Baselines

We first briefly introduce the baseline methods we consider and proceed to our proposed method.

#### 3.1.1 Minimum risk training

Minimum risk training (MRT) (Shen et al., 2016) is one of the most representative RL practices for conditional text generation. It is a slightly altered variation of the REINFORCE algorithm (Williams, 1992), or Monte Carlo policy gradient. Initially adopted in NLP for machine translation task, MRT uses sequences sampled from the model to estimate the gradient of the expected cumulative reward to be obtained under the current policy. It minimizes the *risk*, or the negative expected reward

$$\mathcal{L} = -\sum_{\mathbf{y} \in \mathbf{Y}} q(\mathbf{y}) u(\mathbf{y}; \mathbf{x}), \quad (2)$$

where $\mathbf{Y}$ is the set of samples generated with beam search, and $q(\mathbf{y})$ is the weight of $\mathbf{y}$, which is computed as the normalized model-assigned likelihood

of $\mathbf{y}$ with tempering. The weight term $q$ is similar to the importance weight in importance sampling, but it differs in that here $q$ values are normalized over the set $\mathbf{Y}$.

#### 3.1.2 BRIO

BRIO (Liu et al., 2022), one of the state-of-the-art models for abstractive summarization, also serves as the baseline for our study.

Contrastive learning has been proven effective at learning embeddings of images and sentences (Chen et al., 2020; Gao et al., 2021), and it has recently gained attention within conditional text generation. BRIO utilizes a contrastive objective between full sequences for rank learning. It assigns higher (length-normalized) probability $f$ to sequences with high utility $u$ using a pairwise contrastive loss

$$\mathcal{L}_c = \sum_{u^i > u^j} \max\left(0, \ f(j) - f(i) + \lambda(j - i)\right), \quad (3)$$

where $i$ and $j$ are the ranks of two samples $\mathbf{y}^i$ and $\mathbf{y}^j$, with $i$ being the higher rank, that is, $u^i := u(\mathbf{y}^i; \mathbf{x}) > u(\mathbf{y}^j; \mathbf{x}) =: u^j$, and $f(i) := (1/T^\alpha) \sum_t \log p_\theta(y_t^i \mid \mathbf{x}, \mathbf{y}_{<t}^i)$ is the length penalized log likelihood of the sequence $\mathbf{y}^i$ with length penalty hyperparameter $\alpha$. $\lambda$ is a hyperparameter for determining the margin which is proportional to the rank difference $j - i$. This contrastive loss is then used in conjunction with the MLE loss (the negative log-likelihood loss for the reference) to train the model:

$$\mathcal{L} = \mathcal{L}_{\text{MLE}} + \gamma \cdot \mathcal{L}_c. \quad (4)$$

### 3.2 Motivation

Our primary goal is to have the ability to evaluate *individual actions*. However, MRT and BRIO can compare only fully generated sequences $\mathbf{y}^1$ and $\mathbf{y}^2$, making it challenging to estimate the values of individual actions or partial sequences $\mathbf{y}_{t_1}^1$ and $\mathbf{y}_{t_2}^2$ within those sequences.

To address this limitation and enable the model to understand the consequences of individual actions, we have the following requirements:

- **R1**: We need a way to measure the desirability of an action or a partial sequence based on sparse sequence-level utility $u$. This would allow us to estimate how desirable a partial sequence is, even with this limited information.

- **R2**: We should be able to determine which actions or partial sequences participate in the model update. When we modify the BRIO objective to focus on partial sequences instead of full sequences, we should identify a subset of partial sequence pairs that contribute to rank learning, to avoid requiring the model to meet an overly ambitious goal of estimating a full order of partial sequences.

By fulfilling these requirements, we can overcome the limitations of existing methods and enable the model to learn the value of individual actions and enhance their estimation of desirability.

### 3.3 Proposed: PPR

We now formally present our proposed method, PPR, that addresses R1 and R2, posed in 3.2, elegantly at once. This is achieved by leveraging the prefix tree $\mathcal{T}$ constructed from the set of sample sequences $\mathbf{Y}$ generated from the model conditioned on the source sequence $\mathbf{x}$.

For **R1**, PPR infers the value $V_{\mathcal{T}}(\mathbf{y}_{\leq t})$ of a partially generated sequence $\mathbf{y}_{\leq t}$ using the prefix tree $\mathcal{T}$ as

$$V_{\mathcal{T}}(\mathbf{y}_{\leq t}) := \max_{\mathbf{y}'_{\leq t} = \mathbf{y}_{\leq t}} u(\mathbf{y}'; \mathbf{x}), \qquad (5)$$

where the max operator ranges over $\mathbf{y}' \in \mathbf{Y}$. In other words, $V_{\mathcal{T}}(\mathbf{y}_{\leq t})$ is the highest utility observed among the sequences sharing the common prefix $\mathbf{y}_{\leq t}$. It serves as the missing ground-truth quality measure for partially generated sequences, just as the utility metric $u$ evaluates fully generated sequences.

The prefix tree structure allows to efficiently enumerate over all sequences with the prefix $\mathbf{y}_{\leq t}$, as they correspond to the leaf nodes of the subtree rooted at the node $\mathbf{y}_{\leq t}$. This way, $V_{\mathcal{T}}(\mathbf{y}_{\leq t})$ can be efficiently determined for any given prefix $\mathbf{y}_{\leq t}$.

As previously explained in Section 1, although with $V_{\mathcal{T}}$ we can measure the desirableness of any prefix of any sequence in $\mathbf{Y}$, it can be an unrealistic goal to learn to make pointwise estimates of $V_{\mathcal{T}}$, which requires the model to decide ordering for arbitrary pairs of partially generated sequences in essence. Rather, we adopt the *pairwise ranking* approach of considering two actions as a pair and encourage the model to find the action that results in a partial sequence with higher $V_{\mathcal{T}}$.

Regarding **R2** and how to construct the set of pairs to present to the model, the prefix tree also

1. Generate outputs and rank with $u()$.

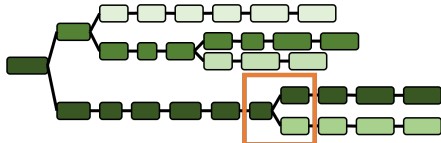

2. Construct a prefix tree from outputs.

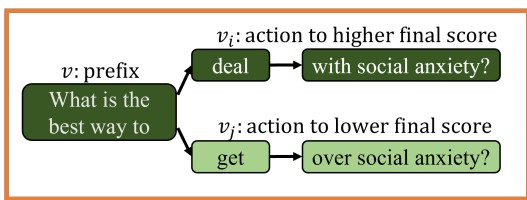

3. Train for $p(v_i|v) - p(v_j|v) > \lambda$ for each branch.

Figure 4: Brief illustration of the PPR training objective. For all 'branching nodes' found from the prefix tree of outputs, PPR applies a max-margin style loss function to assign higher probabilities to actions that will eventually lead to higher sequence-level utility scores.

provides a natural solution. We select the *branching nodes* in the prefix tree and compare the actions corresponding to each branch.

The branching nodes in the prefix tree $\mathcal{T}$ are well-suited for the task of selecting a *feasible* and *necessary* set of pairs to maximize the benefit of learning to rank. This is because:

- We present the model with pairs of partial sequences that share all but the last token. This requirement allows the model to assess the impact of a one-step move, which is more *feasible* compared to distinguishing pairs of partial sequences with much longer chains of differing actions.

- The branches in the prefix tree were kept during the beam search because they were assigned sufficient probability mass by the model. Therefore, the model should be capable of distinguishing these challenging pairs in order to perform well.

We now provide a detailed description of how the pairwise ranking objective is obtained.

As illustrated in Figure 4, we begin by constructing a prefix tree $\mathcal{T}$ using the outputs generated with

the beam search.[3] In this tree, each node $v$ represents a prefix of at least one sample sequence in $\mathbf{Y}$, and each edge corresponds to a token.

Next, we determine the rank of each token $w_i$ following the same prefix $v$ in accordance to $max(V_{\mathcal{T}}(\overline{v_i}))$, the maximum possible utility when the generation starting from $v_i = \overline{vw_i}$ is complete. For all branching node $v$ in $\mathcal{T}$ with $\deg(v) > 1$ children nodes, we rank the associated edges $w_i$ and next states $v_i$ by $V_{\mathcal{T}}(v_i)$. As a result, selecting $w_1$ and transitioning from $v$ to $v_1$ leads to the best outcome, while choosing $w_{\deg(v)}$ is the worst choice possible.

We provide a higher reward for assigning a higher probability to $w_i$ as the next token compared to $w_j$ if $i < j$. The resulting pairwise ranking loss of PPR is in the form

$$\mathcal{L}_c = \sum_{\deg(v) \geq 2} \sum_{(i,j) \in P_v} \max(0, p'(v_j) - p'(v_i) + \lambda),$$

(6)

where $p'(v_i) := p_\theta(w_i \,|\, v, \mathbf{x})$ is the probability of choosing the token $w_i$ conditioned on the source $\mathbf{x}$ and the prefix $v$, and $P_v$ is the set of pairs of indices that will participate in the comparison. For nodes with more than two children, $\deg(v) > 2$, there are several possibilities for selecting which pairs to contrast, $P_v$. We considered the following candidates,

- Dominate:
  $P_v = \{(1,2), (1,3), \cdots, (1,N)\}$,

- Adjacent pairs:
  $P_v = \{(1,2), (2,3), \cdots, (N-1,N)\}$,

- All pairs:
  $P_v = \{(i,j) \,|\, 1 \leq i < j \leq N\}$,

where $N = \deg(v)$, and proceeded with *Dominate* which performed well in our preliminary experiments. In practice, the difference was almost indistinguishable as most of the branching nodes had two children.

Finally, we add pseudo-reference loss $\mathcal{L}'_{\text{MLE}}$, where the sequence with the highest utility in $\mathbf{Y}$ serves as another reference sequence to obtain the following training objective for PPR:

$$\mathcal{L} = \mathcal{L}_{\text{MLE}} + \mathcal{L}'_{\text{MLE}} + \gamma \cdot \mathcal{L}_c \qquad (7)$$

$\mathcal{L}'_{\text{MLE}}$ encourages the model to prefer high-quality sequences it has generated on sequence-level, as elaborated in Section 5.1.

---

[3]A prefix tree is generated on-fly per each input sequences, as described in Equation 7.

# 4  Experiments

## 4.1  Experiment settings

### 4.1.1  Task

As a main evaluation task, we focus on paraphrase generation to evaluate the effectiveness of the proposed PPR for conditional text generation, we experiment on the paraphrase generation task. A paraphrase $\mathbf{y}$ of the source $\mathbf{x}$ exhibits high utility $u(\mathbf{y}; \mathbf{x})$ if it (1) preserves the meaning $\mathbf{x}$ conveys, (2) is a fluent sentence, and (3) deviates lexically and structurally from $\mathbf{x}$, discouraging the copying behaviors $\mathbf{y} = \mathbf{x}$ as a low utility. Regarding generalization to other tasks, please see Subsection 5.4.

### 4.1.2  Architecture and training details

We utilize an encoder-decoder transformer architecture as the policy network. Specifically, we start from the pretrained checkpoints of BART (Lewis et al., 2020) and T5 (Raffel et al., 2020). We provide the detailed hyperparmeter settings and other relevant configuration in Appendix A.

### 4.1.3  Datasets

We use two popular benchmarks on paraphrase generation, namely **QQP-Pos** (Chen et al., 2017) and **MSCOCO** (Lin et al., 2014) for training and evaluation. QQP-Pos is a subset of Quora Question Pairs (QQP) paraphrase identification corpus, where only positive paraphrase pairs are selected. Throughout this paper, we denote QQP-Pos as QQP for simplicity. MSCOCO is a dataset for image captioning, and two captions that describe the same image are treated as a paraphrase pair. For both datasets, we use the standard data splits.

Roughly, QQP consists of more semantically and lexically similar paraphrases than MSCOCO. The self-BLEU (BLEU score between two paraphrases) measured from the test split is 18.35 in QQP, while only 2.46 in MSCOCO.

### 4.1.4  Metrics

While numerous metrics have been proposed for evaluating paraphrase generation automatically (Cao and Wan, 2020; Shen et al., 2022; Nighojkar and Licato, 2021), we choose **BERT-iBLEU** (Niu et al., 2021), which is a reference-free metric capturing both semantic similarity (using BERT-score) and lexical dissimilarity (using iBLEU). It is reported to outperform most of the source-only and reference-only metrics in human judgement

correlation with low computation overhead (Shen et al., 2022; Nighojkar and Licato, 2021).

BERT-iBLEU score is calculated as a weighted harmonic mean of the BERT-score and the iBLEU score between the source and the output. Following the original paper, we set the weight hyperparameter $\beta = 4$ to match the scale of the two metrics. Outputs that are nearly identical to the source sentence result in low iBLEU, and thus penalized in terms of BERT-iBLEU score.

$$\text{BERT-iBLEU} = \frac{\beta + 1}{\frac{\beta}{\text{BERT-score}} + \frac{1}{1 - \text{BLEU}}} \quad (8)$$

We used BERT-iBLEU as the utility $u(\cdot)$, that is, both for evaluating the model performance and for rewarding the model during training.

## 4.2 Results

The result on two base architectures (BART, T5) and two paraphrase generation benchmarks (QQP, MSCOCO) is presented in Table 1. We have evaluated the models with different decoding strategies, namely (1) unbiased sampling and (2) beam search with varying beam sizes (1, 2, 4, 8 and 16) where beam size of 1 is precisely the greedy decoding. The full results, including scores for all beam sizes along with the standard deviation can be found in Appendix C; also, the score-beam size plot can be found in Figure 5.

Our proposed PPR consistently outperforms all baselines in all (model, dataset) pairs. We also present the interpretation of these results through the lens of beam monotonicty and resilience introduced in Section 1.

First, regarding the beam monotonicity, MLE and MRT failing to benefit from increasing beam size and thus violating it indicates a poor correlation or calibration between the likelihood assigned by these models and the target utility score. This suggests that the values of partial sequences implicitly maintained by these models do not align well with the final desired outcome.

Second, for beam resilience, while BRIO does show performance improvement with larger beam sizes, its significant performance decline with smaller beam sizes or unbiased sampling, failing to achieve resilience indicates a failure to consistently estimate the values of partial sequences. This suggests that BRIO's high-ranking candidates are not trustworthy in early decoding stages, as we further anayze in Section 5.2.

| QQP dataset | | | | |
|---|---|---|---|---|
| Model | | Sampling | Beam=1 | Beam=16 |
| BART | MLE | 75.86 | 67.92 | 68.64 |
| | MRT | 78.12 | 73.64 | 77.59 |
| | BRIO | 80.04 | 81.49 | 95.39 |
| | PPR | **93.62** | **96.65** | **96.79** |
| T5 | MLE | 71.81 | 52.16 | 45.57 |
| | MRT | 72.20 | 67.57 | 65.06 |
| | BRIO | 80.88 | 84.84 | 95.27 |
| | PPR | **92.14** | **95.31** | **95.64** |
| Reference | | | 79.28 | |
| MSCOCO dataset | | | | |
| Model | | Sampling | Beam=1 | Beam=16 |
| BART | MLE | 73.72 | 76.40 | 75.32 |
| | MRT | 73.98 | 77.10 | 76.20 |
| | BRIO | 75.91 | 81.89 | 96.62 |
| | PPR | **90.78** | **96.03** | **97.00** |
| T5 | MLE | 73.72 | 74.93 | 71.22 |
| | MRT | 68.94 | 72.13 | 71.70 |
| | BRIO | 74.25 | 81.74 | 95.87 |
| | PPR | **91.39** | **95.96** | **96.10** |
| Reference | | | 73.61 | |

Table 1: BERT-iBLEU scores on paraphrase generation with different decoding strategies: unbiased sampling, greedy ('Beam=1') and beam search ('Beam=16'). The row 'Reference' denotes the average score of reference paraphrases.

| Model | | QQP | MSCOCO |
|---|---|---|---|
| BART | MLE | 12.41 | 2.26 |
| | MRT | 2.31 | 1.89 |
| | BRIO | 0.07 | 0.00 |
| | PPR | 0.03 | 0.00 |
| T5 | MLE | 38.56 | 6.47 |
| | MRT | 36.02 | 5.66 |
| | BRIO | 0.01 | 0.00 |
| | PPR | 0.02 | 0.00 |

Table 2: The portion of model output exactly copying the source sequence (%). MLE clearly fails to learn that copying is not desirable.

Our distinction is that PPR exhibits a nearly flat curve in Figure 5, indicating that the values of partial sequences estimated by our model align well with the actual final rewards obtained by continuing generation from those sequences. Although PPR was not explicitly designed for fully stochastic policy scenarios, where unbiased sampling is used for decoding, it demonstrates significant performance improvement in such circumstances.

Now we examine the failure of MLE in more detail. In addition to its BERT-iBLEU scores generally ranking the lowest, Table 2 highlights its another weakness: the exact copy rate. This metric represents the percentage of model outputs that are exactly the same as the input sequence. Despite the training data consisting of (source, reference) pairs that explicitly discourage exact copying as a

| Model | T5 | |
|---|---|---|
| Dataset | QQP | MSCOCO |
| $\mathcal{L}_{\text{MLE}}$ | 45.57 | 71.22 |
| $\mathcal{L}'_{\text{MLE}}$ | 77.41 | 91.24 |
| $\mathcal{L}_c$ | 82.48 | 96.21 |
| $\mathcal{L}_{\text{MLE}} + \mathcal{L}'_{\text{MLE}}$ | 43.48 | 70.37 |
| $\mathcal{L}_c + \mathcal{L}_{\text{MLE}}$ | 94.50 | 95.44 |
| $\mathcal{L}_c + \mathcal{L}'_{\text{MLE}}$ | 95.50 | 96.42 |
| All | 95.64 | 96.10 |

Table 3: Ablation results for three training objectives in Equation 7, with beam size of 16 at inference time. Results are from single-run experiments.

desirable paraphrase, MLE training fails to capture this aspect from the data.

From an RL perspective, MLE training can be seen as behavioral cloning, where an expert demonstrates desirable sequences of actions, and the agent learns by simply imitating those actions. Our results indicate that, when the desired goal is complex and multifaceted, simply following good paths leads to simple copying behaviors, which is worsened in QQP, which contains a larger proportion of similar (quantitatively, high self-BLEU) paraphrases compared to MSCOCO.

## 5 Analysis

We provide in-depth analysis of the behavior of our model and baselines.

### 5.1 Ablation study

Table 3 presents the ablation study results, where we compare the different composition of training objective components listed in Equation 7.

First, the importance of the contrastive objective $\mathcal{L}_c$ is clearly demonstrated, as models trained with $\mathcal{L}_c$ generally outperform ones trained without it. Furthermore, alongside the performance gain in sufficient beam sizes, $\mathcal{L}_c$ alone can achieve beam monotonicity and resilience, *i.e.* model's robustness with varying beam sizes during inference as illustrated in Table 7.

Also, the pseudo-reference loss plays a key role in promoting the top-ranked candidate based on sequence-level utility. More detailed results regarding the pseudo-reference loss are available in Table 8. Finally, training without the reference loss, or reference-free in other words, led to a small drop in score for the QQP task but a gain for MSCOCO. This suggests that the reference sequences in QQP were better aligned with the source sequences, providing more helpful guidance during training.

### 5.2 Step-wise analysis

We analyze the model's beam resilience in a detailed, step-wise fashion. First, we select the top-ranked sequence with a beam size of 16 (referred as *top beam*). For each generation step, we computed the reciprocal rank of the chosen token within the entire vocabulary and averaged it over all sequences with a length of 10 (including EOS), which was the most common length observed.

Figure 6 illustrates an interesting finding regarding the behavior of baseline methods compared to PPR. During the early to middle steps, baselines often do not consider the prefix of the top beam sequence as desirable. However, as they progress, they change their decisions and favor the prefix. In contrast, PPR consistently achieves high token MRR scores across all time steps and in all cases. This implies that, even with greedily following the best actions, PPR is able to generate a sequence that closely resembles the one obtained through beam search.

This observation has two important implications. Firstly, it directly explains PPR's success in achieving beam resilience during greedy decoding. Secondly, it strongly supports the effectiveness of PPR's pairwise ranking loss for partial sequences in enabling the model to consistently and accurately estimate the values of partial sequences, eliminating the need for looking ahead possible continuations as in beam search, to obtain sequences with high utility.

### 5.3 Additional metrics

To prove robustness of the proposed PPR against different metrics, we propose a variant of BERT-iBLEU, **BiP** (BERT-iBLEU-PPL), which includes an additional perplexity term for measuring fluency of the generated paraphrase. When an output $\mathbf{y}$'s perplexity is higher than the source sentence $\mathbf{x}$ in a scale of $k$, the score is proportionally penalized.

$$\text{BiP} = \text{BERT-iBLEU} \cdot \min\left(1, \frac{k \cdot \text{PPL}(\mathbf{x})}{\text{PPL}(\mathbf{y})}\right) \quad (9)$$

Automatic evaluation results for models trained with BiP as utility are provided in Table 4, where PPR clearly demonstrates beam monotoniticy and resilience together with high performance. For this purpose we have also reported results obtained by evaluating with metrics other than BiP, including ParaScore (Shen et al., 2022), which was designed to better align with human judgment.

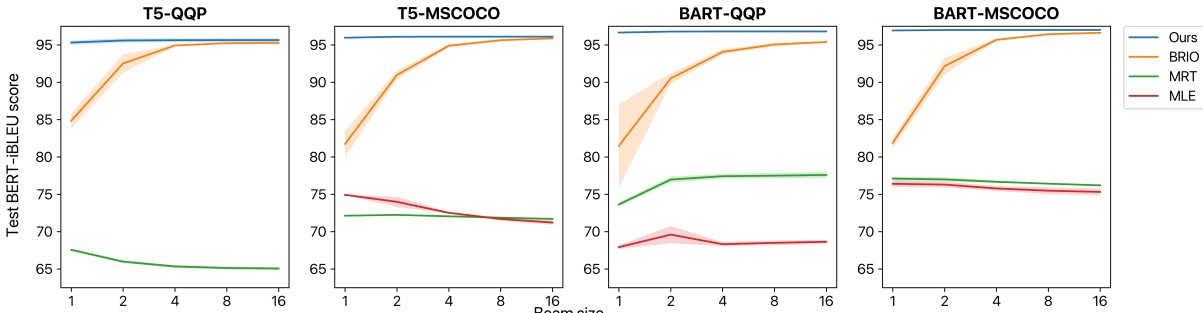

Figure 5: Test BERT-iBLEU score obtained from beam search decoding with varying beam sizes, 1 to 16. Shaded area represents standard deviation. The MLE curve for T5-QQP is excluded due to y-axis coverage.

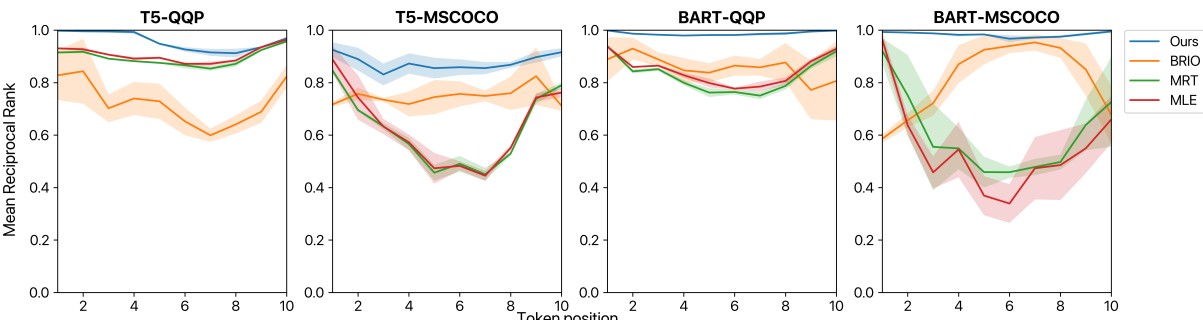

Figure 6: The token mean reciprocal rank (MRR) for the top ranked sequence over time steps. The curve from PPR consistently remains at the top, maintaining values close to 1 as opposed to baseline methods displaying their strong dependence on the tolerance provided by beam search. Shaded area represents standard deviation.

| BiP | | | | |
|---|---|---|---|---|
| | Model | Sampling | Beam=1 | Beam=16 |
| | MLE | 54.01 | 69.47 | 67.28 |
| T5 | BRIO | 51.53 | 81.11 | **88.84** |
| | PPR | **74.84** | **87.79** | 88.57 |
| **BERT-iBLEU** | | | | |
| | Model | Sampling | Beam=1 | Beam=16 |
| | MLE | 73.72 | 74.93 | 71.22 |
| T5 | BRIO | 75.12 | 85.30 | 90.52 |
| | PPR | **85.37** | **90.13** | **90.55** |
| **Parascore** | | | | |
| | Model | Sampling | Beam=1 | Beam=16 |
| | MLE | 32.55 | 17.35 | 3.00 |
| T5 | BRIO | 72.91 | 85.46 | 89.91 |
| | PPR | **84.52** | **89.89** | **89.98** |

Table 4: Automatic evaluation results of T5 models trained on BiP and MSCOCO dataset. We also present cross-evaluation results to show the effectiveness of training with PPR using BiP metric.

Also, we report human evaluation results for different models. We performed example-wise comparison between randomly chosen outputs of PPR and BRIO/MLE trained models (generated by beam search with beam size of 16), asking the annotators to decide which one is a better paraphrase of the given source sentence, or if they are comparable in terms of faithfulness, lexical divergence and

fluency. As a result, PPR won 24%/58% of the case and lost 18%/12% of the case compared to BRIO and MLE, respectively. Qualitatively, our strength was found with respect to faithfulness out of the three main requirements. Examples of generated samples and more details on (automatic) evaluation results can be found in Appendix C and B.

### 5.4 Generalizing PPR to constrained MT

In addition to the paraphrase generation task mainly considered in this paper, we also conducted experiments on lexically constrained machine translation, where the goal is to output a hypothesis given a source sentence to translate plus lexical constraints, which are series of chunks of tokens that must appear on the generated sentence. This dual goal of generating plausible and constraints-satisfying translation has been assessed with two metrics, namely BLEU and copying success rate (CSR[4]) (Wang et al., 2022; Chen et al., 2021).

Table 5 shows the results on constrained machine translation task on the WMT16 En-Ro benchmark. BRIO and PPR used BLEU·CSR, the product of sentence-level BLEU score and CSR as the util-

---

[4]Sometimes also denoted as EM (exact match).

| BLEU | | | |
|---|---|---|---|
| Model | Sampling | Beam=1 | Beam=16 |
| MLE | 22.92 | 27.01 | 27.79 |
| Marian BRIO | 23.67 | 26.37 | 27.21 |
| PPR | **24.70** | **27.86** | **28.46** |
| CSR | | | |
| Model | Sampling | Beam=1 | Beam=16 |
| MLE | 81.96 | 83.97 | 84.99 |
| Marian BRIO | 89.48 | 90.29 | 92.33 |
| PPR | **89.98** | **92.33** | **93.23** |
| BLEU·CSR | | | |
| Model | Sampling | Beam=1 | Beam=16 |
| MLE | 20.01 | 23.97 | 24.87 |
| Marian BRIO | 21.76 | 24.81 | 25.59 |
| PPR | **23.12** | **26.36** | **27.14** |

Table 5: BLEU, CSR (Copying Success Rate), and BLEU·CSR scores from the constrained machine translation task in WMT16 En-Ro benchmark.

ity for training. It is shown that our proposed PPR effectively pursues the dual goal of generating plausible translation containing the required constraints. Further experimental details and complete results can be found in Appendix A and C.

# 6 Related work

This section describes previous work addressing the misbehavior often observed from conditional text generation. Null sequences, repeated $n$-grams and gibberish can be generated as the model assigns high probability to them under certain circumstances (Stahlberg and Byrne, 2019; Holtzman et al., 2020, *inter alia*). We categorize efforts based on whether they attribute these to the decoding method (Subsection 6.1) or to the model's policy (Subsection 6.2).

## 6.1 Decoding methods

Some attribute the degeneration problem to the decoding method itself, as Meister et al. (2020) examined the inductive bias of beam search. Alternative decoding methods to mitigate ill-formed outputs, such as sampling-based approaches (Fan et al., 2018; Holtzman et al., 2020), regularized beam search (Yang et al., 2018; Meister et al., 2020), and constrained decoding (Kajiwara, 2019; Niu et al., 2021) have been propsoed. In paraphrase generation, constrained decoding has been explored to discourage direct copying of the input.

These methods can be seen as ad-hoc solutions without addressing the root cause, unreliable value estimation, as we argue with beam monotonicity and resilience in Section 4.2.

## 6.2 Reward shaping

On the other hand, *reward shaping* can be employed to directly address the value prediction problem, either by (1) providing additional reward on top of the original one to nudge the model towards accomplishing *subgoals*, or (2) distributing the reward over actions through potential-based method. An example of the former is Chan et al. (2019), where generating each keyphrase is rewarded by using recall as the utility in keyphrase generation.

As an example of the latter, in the potential-based method proposed by Bahdanau et al. (2017) for training actor-critic model on neural machine translation task, a partial reward at each time step is defined as the difference between the BLEU scores of the prefix of length $t$ and $t-1$ with respect to the entire reference sentence:

$$R_t = u_p(\mathbf{y}_{\leq t}; \mathbf{x}) - u_p(\mathbf{y}_{<t}; \mathbf{x}), \qquad (10)$$

where the potential, or the utility of a partial sequence $u_p(\mathbf{y}_{\leq t}; \mathbf{x}) = \tilde{u}_p(\mathbf{y}_{\leq t}; \mathbf{x}, \hat{\mathbf{y}})$ is the BLEU score of the partial sequence $\mathbf{y}_{\leq t}$ with respect to the whole reference sequence $\hat{\mathbf{y}}$.

However, it is hard to define such a task-specific potential over all states so that the resulting rewards would faithfully reflect the desirableness of each action, which explains a marginal gain from the above partial reward. In contrast, our design of PPR allows to provide partial rewards directly based on the values of the resulting states.

# 7 Conclusion

We proposed PPR (Pairwise Partial Reward), a novel approach to reward shaping through providing partial rewards in reinforcement learning for conditional text generation. Our method utilizes prefix tree constructed from the set of sample sequences to learn to rank between action pairs, supporting any sequence-level utility metrics $u$ for optimization. By considering samples collectively, our approach enables the model to reflect on its past decisions and immediately determine which actions led to favorable outcomes. This allows PPR to effectively guide the model in estimating values and selecting desirable actions during generation. Experimental results demonstrate that our model outperforms baselines, especially with smaller beam sizes or unbiased sampling, indicating a significant improvements in assessing the values of partial sequences and enhancing language models as stochastic policies.

## Acknowledgements

This work was supported by Institute of Information & Communications Technology Planning & Evaluation (IITP) grant funded by the Korean government (MSIT) (No. 2022-0-00077, AI Technology Development for Commonsense Extraction, Reasoning, and Inference from Heterogeneous Data).

## Limitations

One limitation of our work is that it incurs exploration (sampling) overhead during training, which is common among reinforcement learning approaches to text generation. With this, PPR may require more time to update the model parameters compared to MLE, but surpasses MLE with a smaller number of updates. The trade-off analysis between two factors needs further exploration.

Another limitation of this paper is that, though our work is readily applicable to any conditional generation tasks with any sequence-level evaluation metric, it still requires an extensive analysis of generalization to more diverse set of tasks with possibly longer target sequences and utility functions with even more complex designs.

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

## A Experimental settings in detail

Here we provide the detailed experimental settings.

### A.1 Paraphrase Generation experiments

We used pretrained T5-small (60M) and BART-base (139M) checkpoints, publicly available at Huggingface Transformers [5] with each corresponding tokenizers (with approximately 30k and 50k vocab size).

We used Adam optimizer with $(\beta_1, \beta_2) = (.9, .999)$ and $\epsilon = 1e - 8$. We also used the linear learning rate scheduler with initial learning rate of 5e-5, batch size of 16, dropout rate of 0.1. We trained models for 5 epochs for both QQP and MSCOCO datasets and then chose the best checkpoint with target metric score (BERT-iBLEU) on the development set.

For methods other than MLE, we used beam search for obtaining on-policy samples at train time. For MRT, we used the fixed beam size of 16. For BRIO and PPR, we experimented with both 16 and 32, and chose the one with better performance on the development set. As a result, for all model and dataset combinations, BRIO performed better with 16 while ours enjoyed the benefit of the larger beam size, 32.

Now we describe method-specific hyperparameter settings for BRIO and PPR. For BRIO, we used the fixed length penalty of $\alpha = 1.0$ for normalizing the sequence likelihood. We have searched over .001, .005, .01, .05, .1 for determining the unit margin $\lambda$, and $\lambda = .1$ was the best for all configurations. For the loss scale hyperparameter we found that $\gamma = 10$ worked the best for most of the cases, over $1, 2, 3, 5, 10$ while $\gamma = 2$ gave the best result for BART on MSCOCO.

For PPR, all the model-dataset combinations shared the same best performing hyperparameter settings, where we have determined the contrastive loss scale hyperparamter $\gamma$ as 10 in the same way as for BRIO, and the margin as $\lambda = 0.5$ among $0.1, 0.5$. As described before, we have put the same weight on the negative log-likelihood loss from the ground-truth reference sequence and the pseudo-reference sequence.

### A.2 Constrained MT experiments

Following previous works, as a fully annotated set of training examples for constrained machine translation task is not widely available, we chose the

approach of extracting constraints from public MT benchmarks for unconstrained MT in which only paired sentences without constraints are available. In order to best align with the actual use case of constrained MT in practice, we leveraged a pretrained MarianMT model trained on En-Ro direction available at Huggingface Transformers[6], and the corresponding tokenizer (with approximately 59k vocab size) to identify the words in the target sentences that the model is least confident with, based on the model's output log probabilities. Note that we used word-level constraints, rather than token-level ones, allowing a series of tokens to constitute a single constraint; the confidence for a word was calculated as the average confidence for tokens in that word. We selected 1-3 words as constraints, proportional to the sentence length. In order for the extracted constraints to be reasonable, we applied language filtering based on language detection tools which is a widely adopted practice in building MT systems based on publicly available data (Ng et al., 2019), removing noisy examples such as those containing German(de) sentences. Among the 610k training examples in WMT16 En-Ro dataset, we only used pairs of which (1) source sentence is classified as En and target sentence is classified as Ro, and (2) both source and target sentences are shorter than 32 tokens, which leaves slightly less than 100k samples. The constraints were appended to the source sentence, where the source and constraints were separated by <eos> token, to be fed as input to the model. Validation and test splits were processed in the same manner.

We initialized the models with the aforementioned pretrained MarianMT checkpoint, which exhibited nearly 40-50% of CSR on its own. The models were trained for 5 epochs with initial learning rate 2e-5, effective batch size 128 and beam size 16. For other method-specific hyperparameters we used the same values as in the main experiments.

## B Examples of generated paraphrases

Table 6 includes generated paraphrase samples from models trained with different metrics and training objectives.

Models trained with MLE often generate a direct copy of the source as shown in Table 2, or omit important details. On the other hand, PPR and BRIO trained with BERT-iBLEU clearly learns to

---

[5] https://github.com/huggingface/transformers

[6] https://huggingface.co/Helsinki-NLP/opus-mt-en-ro

exploit the design of BERT-iBLEU, which allows copying up to 3-grams from the source to maintain high semantic similarity captured by BERTscore yet still avoiding being penalized by BLEU score.

The difference between models trained on BRIO and PPR is best shown in BiP examples. BRIO often leads to generating false information that is not faithful to the content of the source, as highlighted in italic. In contrast, PPR effectively retains, or at least does not flip or change the meaning of important keywords present in the source sentence through directly contrasting token-level actions given the prefix, leading to better choices in critical steps. This showcases our method's effectiveness in token-level credit assignment in autoregressive conditional text generation.

| BERT-iBLEU | |
|---|---|
| Model | Example |
| Input | An office cubicle with four different types of computers. |
| MLE | An office cubicle with four different types of computers. |
| BRIO | An office cubicle,with 4 different computers types. |
| PPR | An office cubicle containing 4 different types computers. |
| Input | A Marine that is looking at his cell phone. |
| MLE | A man is looking at his cell phone. |
| BRIO | A Marine that's looking at his cell-phone. |
| PPR | A Marine who is looking on his cellphone. |
| BiP | |
| Model | Example |
| Input | A hotel has a *small* tv on the dresser. |
| MLE | A tv on a dresser in a hotel room. |
| BRIO | A hotel is equipped with a *large* TV on top of the dresser. |
| PPR | A hotel features a tv on top of the dresser. |
| Input | Two *yellow* fruits hanging on branches full of leaves. |
| MLE | Two fruits are hanging on the branches of a tree. |
| BRIO | Two *orange* fruits are hanging on a branch full of leaves. |
| PPR | Two *yellow* fruits hang on a branch full of leaves. |

Table 6: Generation examples from models trained with BERT-iBLEU and BiP on MSCOCO dataset with T5 as the base architecture.

# C Complete results

Table 9 shows the results for paraphrasing in more detail including the standard deviation of each reported figure. Substantially improved performance with oracle reranker shows that MLE and MRT failing to achieve beam monotonicity is largely due to that their failing to align probability and target utility.

In addition, Table 10 displays the complete result for models trained in BiP. Strong performance gain is achieved with PPR compared to its counterparts, especially with unbiased sampling and beam search decoding with small beam sizes.

Finally, Table 11 shows the full results on lexically constrained machine translation task, where PPR outperforms others not only in the given metric for training, BRIO·CSR, but also in its individual components (BRIO and CSR). While models trained using MLE or BRIO objectives tend to frequently violate beam monotonicity, PPR can generally take advantage of the increased beam size.

| Model-Dataset | Sampling | Beam=1 | Beam=2 | Beam=4 | Beam=8 | Beam=16 | Oracle |
|---|---|---|---|---|---|---|---|
| T5-MSCOCO | 92.21 | 96.02 | 96.18 | 96.21 | 96.21 | 96.21 | 97.29 |

Table 7: BERT-iBLEU scores only trained with $\mathcal{L}_c$, without any NLL loss($\mathcal{L}_{\mathrm{MLE}}, \mathcal{L}'_{\mathrm{MLE}}$).

| Model-Dataset | Sampling | Beam=1 | Beam=2 | Beam=4 | Beam=8 | Beam=16 | Oracle |
|---|---|---|---|---|---|---|---|
| BART-QQP | $90.02_{\pm.12}$ | $96.72_{\pm.06}$ | $96.77_{\pm.03}$ | $96.78_{\pm.03}$ | $96.78_{\pm.03}$ | $96.79_{\pm.04}$ | $97.34_{\pm.04}$ |
| BART-MSCOCO | $82.75_{\pm.16}$ | $96.31_{\pm.08}$ | $96.65_{\pm.08}$ | $96.68_{\pm.07}$ | $96.69_{\pm.07}$ | $96.69_{\pm.07}$ | $97.33_{\pm.08}$ |
| T5-QQP | $88.03_{\pm.03}$ | $94.24_{\pm.06}$ | $94.53_{\pm.07}$ | $94.59_{\pm.08}$ | $94.59_{\pm.08}$ | $94.59_{\pm.08}$ | $96.34_{\pm.06}$ |
| T5-MSCOCO | $84.77_{\pm.03}$ | $95.13_{\pm.06}$ | $95.34_{\pm.02}$ | $95.42_{\pm.02}$ | $95.43_{\pm.01}$ | $95.44_{\pm.01}$ | $96.71_{\pm.03}$ |

Table 8: BERT-iBLEU scores from PPR without the pseudo-reference loss. Still, shrinking the beam size has little effect on the performance.

| | Model | Sampling | Beam=1 | Beam=2 | Beam=4 | Beam=8 | Beam=16 | Oracle |
|---|---|---|---|---|---|---|---|---|
| \multicolumn{9}{c}{QQP dataset} | | | | | | | | |
| BART | MLE | $75.86_{\pm.04}$ | $67.92_{\pm.24}$ | $69.60_{\pm1.15}$ | $68.32_{\pm.26}$ | $68.50_{\pm.32}$ | $68.64_{\pm.26}$ | $88.61_{\pm.05}$ |
| | MRT | $78.12_{\pm.13}$ | $73.64_{\pm.27}$ | $76.97_{\pm.40}$ | $77.42_{\pm.32}$ | $77.49_{\pm.37}$ | $77.59_{\pm.39}$ | $88.75_{\pm.19}$ |
| | BRIO | $80.04_{\pm1.83}$ | $81.49_{\pm5.54}$ | $90.51_{\pm.71}$ | $94.06_{\pm.34}$ | $95.04_{\pm.22}$ | $95.39_{\pm.13}$ | $96.16_{\pm.27}$ |
| | PPR | $\mathbf{93.62}_{\pm.08}$ | $\mathbf{96.65}_{\pm.12}$ | $\mathbf{96.78}_{\pm.06}$ | $\mathbf{96.80}_{\pm.06}$ | $\mathbf{96.80}_{\pm.06}$ | $\mathbf{96.79}_{\pm.05}$ | $\mathbf{97.23}_{\pm.07}$ |
| T5 | MLE | $71.81_{\pm.09}$ | $52.16_{\pm.50}$ | $47.81_{\pm.43}$ | $46.23_{\pm.34}$ | $45.72_{\pm.38}$ | $45.57_{\pm.34}$ | $88.60_{\pm.03}$ |
| | MRT | $72.20_{\pm.05}$ | $67.57_{\pm.16}$ | $65.98_{\pm.19}$ | $65.34_{\pm.18}$ | $65.13_{\pm.18}$ | $65.06_{\pm.19}$ | $81.71_{\pm.02}$ |
| | BRIO | $80.88_{\pm1.16}$ | $84.84_{\pm1.03}$ | $92.51_{\pm1.23}$ | $94.92_{\pm.11}$ | $95.23_{\pm.04}$ | $95.27_{\pm.13}$ | $96.51_{\pm.05}$ |
| | PPR | $\mathbf{92.14}_{\pm.40}$ | $\mathbf{95.31}_{\pm.29}$ | $\mathbf{95.59}_{\pm.28}$ | $\mathbf{95.63}_{\pm.26}$ | $\mathbf{95.64}_{\pm.26}^*$ | $\mathbf{95.64}_{\pm.26}^*$ | $\mathbf{97.00}_{\pm.08}$ |
| \multicolumn{2}{c}{Reference} | | | | 79.28 | | | | |
| \multicolumn{9}{c}{MSCOCO dataset} | | | | | | | | |
| BART | MLE | $73.72_{\pm.36}$ | $76.40_{\pm.44}$ | $76.30_{\pm.37}$ | $75.78_{\pm.33}$ | $75.48_{\pm.45}$ | $75.32_{\pm.44}$ | $84.33_{\pm.01}$ |
| | MRT | $73.98_{\pm.30}$ | $77.10_{\pm.26}$ | $76.99_{\pm.30}$ | $76.67_{\pm.17}$ | $76.42_{\pm.05}$ | $76.20_{\pm.11}$ | $84.57_{\pm.32}$ |
| | BRIO | $75.91_{\pm.08}$ | $81.89_{\pm.66}$ | $92.15_{\pm1.11}$ | $95.68_{\pm.16}$ | $96.43_{\pm.06}$ | $96.62_{\pm.03}$ | $96.85_{\pm.02}$ |
| | PPR | $\mathbf{90.78}_{\pm.12}$ | $\mathbf{96.93}_{\pm.10}$ | $\mathbf{96.99}_{\pm.10}$ | $\mathbf{97.00}_{\pm.10}$ | $\mathbf{97.00}_{\pm.10}$ | $\mathbf{97.00}_{\pm.10}$ | $\mathbf{97.57}_{\pm.08}$ |
| T5 | MLE | $73.72_{\pm.11}$ | $74.93_{\pm.02}$ | $73.99_{\pm.68}$ | $72.52_{\pm.11}$ | $71.69_{\pm.21}$ | $71.22_{\pm.30}$ | $85.45_{\pm.08}$ |
| | MRT | $68.94_{\pm.06}$ | $72.13_{\pm.09}$ | $72.24_{\pm.08}$ | $72.05_{\pm.02}$ | $71.87_{\pm.10}$ | $71.70_{\pm.10}$ | $78.96_{\pm.06}$ |
| | BRIO | $74.25_{\pm.50}$ | $81.74_{\pm1.74}$ | $90.96_{\pm.58}$ | $94.88_{\pm.22}$ | $95.63_{\pm.14}$ | $95.87_{\pm.11}$ | $96.37_{\pm.10}$ |
| | PPR | $\mathbf{91.39}_{\pm.12}$ | $\mathbf{95.96}_{\pm.02}$ | $\mathbf{96.09}_{\pm.02}$ | $\mathbf{96.10}_{\pm.02}$ | $\mathbf{96.10}_{\pm.02}$ | $\mathbf{96.10}_{\pm.02}$ | $\mathbf{97.16}_{\pm.02}$ |
| \multicolumn{2}{c}{Reference} | | | | 73.61 | | | | |

Table 9: BERT-iBLEU scores on paraphrase generation with different decoding strategies, with standard deviation. The column 'Oracle' denotes the average score of the sequence with the highest score in each set of candidates generated with beam search (of beam size 16), that is, the score obtained with oracle reranking. Our model not only outperforms baselines in all cases but also retains the performance with greedy decoding or unbiased sampling compared to applying beam search with beam size 16, which clearly demonstrates its excelling at estimating the values of partial sequences. All values are averaged over 3 runs with different random seeds. Boldfaced numbers indicate statistical significance under $p < .05$, while boldfaced with stars indicate $p < .07$.

| BiP | | | | | | | |
|---|---|---|---|---|---|---|---|
| | Model | Sampling | Beam=1 | Beam=2 | Beam=4 | Beam=8 | Beam=16 | Oracle |
| | MLE | 54.01 | 69.47 | 68.40 | 68.36 | 67.77 | 67.28 | 84.03 |
| T5 | BRIO | 51.53 | 81.11 | 86.44 | 88.17 | **88.60** | **88.84** | 92.24 |
| | PPR | **74.84** | **87.79** | **88.37** | **88.49** | 88.57 | 88.57 | **92.58** |

| BERT-iBLEU | | | | | | | |
|---|---|---|---|---|---|---|---|
| | Model | Sampling | Beam=1 | Beam=2 | Beam=4 | Beam=8 | Beam=16 | Oracle |
| | MLE | 73.72 | 74.93 | 73.99 | 72.52 | 71.69 | 71.22 | 85.45 |
| T5 | BRIO | 75.12 | 85.30 | 88.77 | 90.04 | 90.40 | 90.52 | 92.73 |
| | PPR | **85.37** | **90.13** | **90.46** | **90.54** | **90.54** | **90.55** | **93.46** |

| Parascore | | | | | | | |
|---|---|---|---|---|---|---|---|
| | Model | Sampling | Beam=1 | Beam=2 | Beam=4 | Beam=8 | Beam=16 | Oracle |
| | MLE | 32.55 | 17.35 | 11.69 | 6.91 | 4.32 | 3.01 | 34.39 |
| T5 | BRIO | 72.91 | 85.46 | 88.39 | 89.50 | 89.81 | 89.91 | 92.06 |
| | PPR | **84.52** | **89.69** | **89.89** | **89.96** | **89.97** | **89.98** | **92.25** |

Table 10: Complete automatic evaluation results of T5 models trained with BiP metric as utility on MSCOCO dataset, evaluated by different metrics. Note that significant performance gain can be achieved in sampling and small beam size settings when trained with PPR, and even when assessed with different evaluation metrics other than the one used as the utility (reward) to train the model, PPR exhibits beam monotonicity and resilience. Values for BRIO and PPR are from single-run experiments.

| BLEU | | | | | | | |
|---|---|---|---|---|---|---|---|
| Model | | Sampling | Beam=1 | Beam=2 | Beam=4 | Beam=8 | Beam=16 |
| | MLE | 22.92 | 27.01 | 27.39 | 27.89 | 27.82 | 27.79 |
| MarianMT | BRIO | 23.67 | 26.37 | 27.01 | 27.36 | 27.36 | 27.21 |
| | PPR | **24.70** | **27.86** | **28.33** | **28.21** | **28.43** | **28.46** |

| CSR | | | | | | | |
|---|---|---|---|---|---|---|---|
| Model | | Sampling | Beam=1 | Beam=2 | Beam=4 | Beam=8 | Beam=16 |
| | MLE | 81.96 | 83.97 | 84.31 | 85.55 | 85.21 | 84.99 |
| MarianMT | BRIO | 89.48 | 90.29 | 92.10 | 92.10 | 92.43 | 92.33 |
| | PPR | **89.98** | **92.33** | **92.66** | **92.89** | **93.00** | **93.23** |

| BLEU·CSR | | | | | | | |
|---|---|---|---|---|---|---|---|
| Model | | Sampling | Beam=1 | Beam=2 | Beam=4 | Beam=8 | Beam=16 |
| | MLE | 20.01 | 23.97 | 24.17 | 25.00 | 24.95 | 24.87 |
| MarianMT | BRIO | 21.76 | 24.81 | 25.49 | 25.69 | 25.73 | 25.59 |
| | PPR | **23.12** | **26.36** | **26.90** | **26.82** | **27.06** | **27.14** |

Table 11: BLEU, CSR(Copying Success Rate%), and BLEU·CSR scores from the constrained machine translation task on WMT16 En-Ro benchmark. BRIO and PPR used BLEU·CSR as utility. Values are from single-run experiments.