# OpenReview forum: "Learning to Rank Generation with Pairwise Partial Rewards"
_EMNLP/2023/Conference — EMNLP 2023 Main_

### Official Review · Reviewer_WfAY · 2023-07-31

**Soundness:** 4

**Excitement:**

4: Strong: This paper deepens the understanding of some phenomenon or lowers the barriers to an existing research direction.

**Missing References:**

The related work section is a little sparse - I suspect there is significant previous work within RL about credit assignment and partial rewards, although I’m not familiar enough with the area to suggest specific papers.

**Paper Topic And Main Contributions:**

The paper proposes a method for improving credit assignment when training text generation models using RL. The method uses a prefix tree to store the rewards assigned to a set of complete candidate generations, and thus to assign rewards to the partial generations. It concentrates on the task of paraphrase generation, with experiments performed on two base models and two datasets. Aside from a major issue with evaluation, this is a well written and interesting paper.

**Questions For The Authors:**

A) Why did the authors not experiment on summarization, given that one of the comparison methods (BRIO) is designed for summarization?

B) Since beam search often leads to a set of fairly similar hypotheses, did the authors experiment with using other sampling methods to generate the candidate outputs used to construct the prefix tree?

**Reasons To Accept:**

The authors propose an elegant solution to an important problem, with potential relevance to RLHF.

The paper is well written and clearly explained.

Experiments are fairly thorough with ablations included. The insight about beam monotonicity and resilience (Figs 2 and 5) is interesting and nicely motivates the proposed method.

**Reasons To Reject:**

The evaluation is based solely on BERT-iBLEU, which is the same metric used as a training objective. It is no surprise therefore that the scores increase! No analysis is performed as to whether the method leads to truly higher quality (and more diverse) paraphrases, or whether it leads to degenerate output that scores highly but is not useful: see e.g. Evaluating Rewards for Question Generation Models, Hosking & Riedel (2019), On Improving Summarization Factual Consistency from Natural Language Feedback, Liu et al. (2023), Reward Gaming in Conditional Text Generation, Pang et al. (2023) and many others for examples of reward gaming. In my view this is a fundamental issue with the paper as it stands and is the main reason for my soundness score.

No human evaluation is included to confirm the automatic results.

Although I recognise that the paper is focussed on sampling from general models, it would be good to see a comparison to other paraphrasing systems.

EDIT: Additional results presented in the rebuttal show that reward gaming does not seem to be an issue. The authors indicate that they will include a human evaluation in any revised version of the paper.

**Reproducibility:**

4: Could mostly reproduce the results, but there may be some variation because of sample variance or minor variations in their interpretation of the protocol or method.

**Reviewer Confidence:**

4: Quite sure. I tried to check the important points carefully. It's unlikely, though conceivable, that I missed something that should affect my ratings.

**Typos Grammar Style And Presentation Improvements:**

Table 4 caption could be more informative. Also, the authors could consider combining Table 4 and 5.

---

> ### Author Rebuttal · Authors · 2023-08-29
>
> **Regarding BERT-iBLEU metric and the 'Related work' section:**
> As the reviewers have pointed out, we are aware that some metrics may not properly correlate with human evaluation or be vulnerable to inadequate outputs, and RL methods using them as rewards can exploit such properties (Pang et al., 2023; [Yan et al., 2023](https://aclanthology.org/2023.acl-long.297)). These are addressed by improved metrics, while the optimization methods are rather orthogonal to the issue (Pang et al., 2023; [Skalse et al., 2022](https://openreview.net/forum?id=yb3HOXO3lX2); [Hao et al., 2023](https://openreview.net/forum?id=1_gypPuWUC3)). We stress our target is the optimization method itself, not the design of the metric, as ours can be applied to any given metric.
>
> However, we acknowledge with the reviewer’s suggestion to discuss these concerns in the prior literature and consider adopting Pang et al. (2023) to penalize gaming via perplexity measure ([Fabre et al., 2021](https://aclanthology.org/2021.eacl-main.180)) for BERT-iBLEU, which we denote as BERT-iBLEU-PPL. Considering another metric also helps us address the reviewer's concern that whether our finding is consistent when training-evaluation metrics differ: Below are additional experiments where we used BERT-iBLEU-PPL for training and evaluated with different metrics, where ours outperforms in all settings.
>
> We will report full results including human evaluation in the revised paper.
>
> #### Evaluation with BERT-iBLEU-PPL
> | Method | Sampling  | Greedy    | BS=2      | 4         |
> |:------:|:---------:|:---------:|:---------:|:---------:|
> | MLE    | 54.01     | 69.47     | 68.40     | 68.36     |
> | BRIO*  | 51.53     | 81.11     | 86.44     | 88.17     |
> | Ours*  | **74.84** | **87.79** | **88.37** | **88.49** |
>
> cf. reference sentences score 55.24
>
> #### Evaluation with BERT-iBLEU
> | Method | Sampling  | Greedy    | BS=2      | 4         |
> |:------:|:---------:|:---------:|:---------:|:---------:|
> | MLE    | 73.72     | 74.93     | 73.99     | 72.52     |
> | BRIO*  | 75.12     | 85.30     | 88.77     | 90.04     |
> | Ours*  | **85.27** | **90.03** | **90.46** | **90.54** |
>
> #### Evaluation with Parascore ([On the Evaluation Metrics for Paraphrase Generation](https://aclanthology.org/2022.emnlp-main.208) (EMNLP 2022))
> | Method | Sampling  | Greedy    | BS=2      | 4         |
> |:------:|:---------:|:---------:|:---------:|:---------:|
> | MLE    | 72.47     | 80.92     | 82.32     | 84.36     |
> | BRIO*  | 72.91     | 85.46     | 88.39     | 89.50     |
> | Ours*  | **84.52** | **89.69** | **89.89** | **89.96** |
>
> The above results are obtained with T5 on MSCOCO dataset, single run experiments with the same set of hyperparameters used in the draft; we will report the average and the standard deviation from multiple runs as we did before in the revised paper.
> Compared to BERT-iBLEU, BERT-iBLEU-PPL additionally takes into account the fluency of the generated paraphrase by penalizing the score when the perplexity of the generated sentence is higher than that of the source sentence.
>
> **Regarding comparison to other paraphrase systems:** We compare our BERT-iBLEU scores with previous paraphrasing systems using pretrained T5 checkpoint and Quora Question Pairs (QQP) dataset, which is the most widely available combination from literature.
> As demonstrated, our claim about the effectiveness from the paper remain unchanged.
>
> | Method | BERT-iBLEU |
> |:------:|:----------:|
> | Ours (as in draft) | 95.64 |
> | Dynamic blocking   | 83.1  |
> | TASSDB      | 77.73 |
> | + RulePG    | 81.97 |
> | Separator   | 80.19 |
> | HRQVAE      | 65.01 |
>
> The numbers are from [Unsupervised Paraphrasing with Pretrained Language Models](https://aclanthology.org/2021.emnlp-main.417) (Niu et al., EMNLP 2021) and [RulePG: Syntactic Rule-enhanced Paraphrase Generation](https://ieeexplore.ieee.org/document/10191854) (Lan et al., IJCNN 2023).
> Note that our result is based on `T5-small` architecture, while other systems have used either `T5-base` or `T5-large`, its larger counterparts.
>
> **Regarding other tasks:** As the reviewer pointed out, as in BRIO, our approach can be straightforwardly adapted for the summarization task. Though we see preliminary promising results, due to the resources required for conducting extensive experiments on summarization benchmarks containing much longer texts, conclusive results will be reported later, either in discussion or the revised draft.
>
> **Regarding sampling methods during training:**
> We have experimented with diverse beam search ([Vijayakumar et al., 2016](https://ojs.aaai.org/index.php/AAAI/article/view/12340)) and unbiased sampling as the sampling method at training time, which was not effective and thus unreported. We susepct two possible causes; (1) prefix trees constructed with these methods tend to branch too early, making it difficult to provide supervisory signals on later steps, and (2) less important actions(tokens) can be over-represented via random sampling or diversification.
>
> **Regarding presentation:** Thanks for the suggestion. We will reorganize the tables for ablation study results in the revised paper.

---

### Official Review · Reviewer_yfPs · 2023-08-05

**Soundness:** 4

**Excitement:**

4: Strong: This paper deepens the understanding of some phenomenon or lowers the barriers to an existing research direction.

**Missing References:**

Please add the citation to the following paper. The following paper also assigns partial rewards to present keyphrases and absent keyphrases for keyphrase generation models:

Neural Keyphrase Generation via Reinforcement Learning with Adaptive Rewards. ACL 2019.

**Paper Topic And Main Contributions:**

This paper aims to address the credit assignment problem of reinforcement learning algorithms in text generation tasks. To this end, it proposes a reward-shaping method to provide partial reward on partially decoded sequences. The idea is to teach a model to distinguish the relative desirability between pairs of decoding actions by utilizing the prefix tree constructed from the sampled sequence.

**Reasons To Accept:**

The motivation of this work is very clear and is well supported by quantitative analysis on the monotonicity and resilience of beam search decoding.

The proposed reward shaping technique for providing partial reward feedback to text generation models is very novel. It is the first work that applies prefix tree for reward shaping in text generation.

The proposed method outperforms previous reinforcement learning and contrastive learning algorithms for text generation.

The automatic evaluations are comprehensive. It extensively analyzes the performance of different algorithms with varying beam sizes. It also conducts a stepwise analysis on the beam resilience of PLMs.


**Reasons To Reject:**

This paper does not conduct human evaluations to verify the quality of the model-generated outputs. The reviewer suggests the authors conduct human evaluation on a random subset of the model outputs.

**Reproducibility:**

5: Could easily reproduce the results.

**Reviewer Confidence:**

4: Quite sure. I tried to check the important points carefully. It's unlikely, though conceivable, that I missed something that should affect my ratings.

---

> ### Author Rebuttal · Authors · 2023-08-29
>
> **Regarding human evaluation:** We agree that human evaluations would help us evaluate whether the reward aligns with human perception of quality and will report in the revised draft.
> Meanwhile, we also want to clarify by pointing out that the design of reward is not our contribution.
> See our answer for reviewer WfAY, on how our work is orthogonal to the choice of reward metrics and our findings remain unchanged for such choice.
>
> **Regarding missing references:** Thanks for the suggestion.
> As the reviewer suggested, the adaptive rewards in Chan et al., 2019 could be seen as providing a partial reward for each keyphrase generation, though the task context differs from ours.
> We will add discussion to the revised paper.

---

### Official Review · Reviewer_4L43 · 2023-08-06

**Soundness:** 4

**Excitement:**

4: Strong: This paper deepens the understanding of some phenomenon or lowers the barriers to an existing research direction.

**Missing References:**

None

**Paper Topic And Main Contributions:**

The supervised learning for conditional text generation suffers from
the exposure bias problem, and reinforcement learning has been widely
used to avoid this problem. However, this method faces the sparse and
delayed reword problem, where the reward can be obtained only when the
generation is finished.

To deal with this problem, this paper presents PPR (Pairwise Partial
Reword), a method for providing partial rewords for intermediate
action tokens. The proposed method first constructs a prefix tree
using sampled sequences, and employs a pairwise ranking objective for
the branching nodes. Then, the model is encouraged to prefer the
branch with a higher expected reward.

The experimental results on the paraphrase generation task demonstrate
the proposed method outperforms several baseline methods.

The proposed method is well motivated, and achieves a substantial gain
over the baseline methods. The extensive analysis supports the
motivation of the proposed method.


**Questions For The Authors:**

- In the ablation study (Section 5.1), how about the score using
  $L_{MLE}$ + $L'_{MLE}$? I want to know the effectiveness of
  $L_c$ alone?


**Reasons To Accept:**

- The proposed method is well motivated.

- The proposed method achieves a substantial gain over the widely-used
  baseline methods.

- The extensive analysis supports the motivation of the proposed
  method.


**Reasons To Reject:**

- It is little difficult to follow the paper, and the paper
  organization is worth reviewing. Figure 4 is the easiest to
  understand, and should be presented in Section 1. Furthermore,
  Figure 2 is difficult to understand as the beginning of the paper
  (especially, the meaning of the baselines (A) and (B) is hard to
  understand). This section should be described after the proposed
  method is described.

- The notations for the proposed method are difficult to
  follow. Please refer to "Typos Grammar Style And Presentation
  Improvements".

- Although mentioned in the Limitations section, the proposed method
  is validated only on the paraphrase generation task. It is not clear
  that the proposed method is effective on the machine translation or
  summarization tasks, where the reinforcement learning is widely
  used.


**Reproducibility:**

3: Could reproduce the results with some difficulty. The settings of parameters are underspecified or subjectively determined; the training/evaluation data are not widely available.

**Reviewer Confidence:**

4: Quite sure. I tried to check the important points carefully. It's unlikely, though conceivable, that I missed something that should affect my ratings.

**Typos Grammar Style And Presentation Improvements:**

- l042: likelihood -> probability

- l215: .. y^2. making .. -> .. y^2, making ..

- l242: address -> addresses

- l242: 3.2 -> Section 3.2

- l295: When is "we begin by constructing a prefix tree.."
  performed? This is performed before training the proposed method
  using a pre-trained model? Please clarify this.

- l298: from the beam search -> with the beam search

- l304: It is hard to follow the presented notations. The subscript
  "1" in w_1 and v_1 and deg(v) in w_{deg(v)} and v_{deg(v)} are hard
  to understand.

- l313: p'(v_{j}) -> log p'(v_{j}), p'(v_{i}) -> log p'(v_{i})

- l320: We have considered -> We consider

- l340: section 5.1 -> Section 5.1

---

> ### Author Rebuttal · Authors · 2023-08-29
>
> **Regarding organization and notations:** Thanks for the suggestion. We will follow those to reorganize our paper.
>
> **Regarding ablation study:** Using only the reference loss $L_{\textrm{MLE}}$ and the pseudo-reference loss $L^\prime_{\textrm{MLE}}$ without the contrastive loss $L_c$ gave nearly the same result as using the reference loss only ("MLE").
> Similarly, excluding both of them incurred minimal changes in performance or training dynamics.
> The following results are from single-run experiments.
>
> | Training objective | Sampling  | Greedy    | BS=2      | 4         |
> |:------:|:---------:|:---------:|:---------:|:---------:|
> | $L_{\textrm{MLE}} + L^\prime_{\textrm{MLE}}$ | 73.69 | 74.78 | 73.26 | 71.64 |
> | $L_c$ | 92.21 | 96.02 | 96.18 | 96.21 |
>
> **Regarding limitation section:** Please see our answer for WfAY.

---

### Meta-Review · Area_Chair_Q3fC · 2023-09-19

**Recommendation:** 5

**Metareview:**

The paper targets the problem of large action space and the delayed reward problem when RL is used for conditional text generation. A thorough evaluation is conducted. A common concern of reviewers is of human evaluation which authors agree to include in the final version of the paper. All the other experiments using different metrics and comparisons with other baselines are provided during rebuttal.

---

### Decision · Program_Chairs · 2023-10-07

**Decision:**

Accept-Main

**Comment:**

The paper targets the problem of large action space and the delayed reward problem when RL is used for conditional text generation. A thorough evaluation is conducted. A common concern of reviewers is of human evaluation which authors agree to include in the final version of the paper. All the other experiments using different metrics and comparisons with other baselines are provided during rebuttal.